# High-Performance Overlay Analysis of Massive Geographic Polygons That Considers Shape Complexity in a Cloud Environment

**Kang Zhao [1], Baoxuan Jin [2],\*, Hong Fan [1], Weiwei Song [3], Sunyu Zhou [3] and Yuanyi Jiang [3]**

[1] State Key Laboratory for Information Engineering in Surveying, Mapping, and Remote Sensing, Wuhan University, 129 Luoyu Road, Wuhan 430079, China

[2] Information Center, Department of Land and Resources of Yunnan Province, Kunming 650504, China

[3] Department of Geoinformation Science, Kunming University of Science and Technology, Kunming 650504, China

\* Correspondence: jbx@yngc.org; Tel.: +86-0871-65747357

**Abstract:** Overlay analysis is a common task in geographic computing that is widely used in geographic information systems, computer graphics, and computer science. With the breakthroughs in Earth observation technologies, particularly the emergence of high-resolution satellite remote-sensing technology, geographic data have demonstrated explosive growth. The overlay analysis of massive and complex geographic data has become a computationally intensive task. Distributed parallel processing in a cloud environment provides an efficient solution to this problem. The cloud computing paradigm represented by Spark has become the standard for massive data processing in the industry and academia due to its large-scale and low-latency characteristics. The cloud computing paradigm has attracted further attention for the purpose of solving the overlay analysis of massive data. These studies mainly focus on how to implement parallel overlay analysis in a cloud computing paradigm but pay less attention to the impact of spatial data graphics complexity on parallel computing efficiency, especially the data skew caused by the difference in the graphic complexity. Geographic polygons often have complex graphical structures, such as many vertices, composite structures including holes and islands. When the Spark paradigm is used to solve the overlay analysis of massive geographic polygons, its calculation efficiency is closely related to factors such as data organization and algorithm design. Considering the influence of the shape complexity of polygons on the performance of overlay analysis, we design and implement a parallel processing algorithm based on the Spark paradigm in this paper. Based on the analysis of the shape complexity of polygons, the overlay analysis speed is improved via reasonable data partition, distributed spatial index, a minimum boundary rectangular filter and other optimization processes, and the high speed and parallel efficiency are maintained.

**Keywords:** overlay analysis; shape complexity; massive data; cloud; parallel computing

---

## 1. Introduction

Overlay analysis is a common geographic computing operation and an important spatial analysis function of geographic information systems (GIS). It is widely used in applications related to spatial computing [1,2]. This operation involves the spatial overlay analysis of different data layers and their attributes in the target area. It connects multiple spatial objects from multiple data sets, creates a new clip data set, and quantitatively analyzes the spatial range and characteristics of the interactions among different types of spatial objects [3]. The development of geospatial science has entered a new stage with the rapid popularization of the global Internet, sensor technologies, and Earth observation technologies. The transformation of a space information service from digital Earth to intelligent Earth has posed

challenges, such as being data-intensive, computationally intensive, and time–space-intensive and high concurrent access [4,5]. Overlay analysis deals with massive data, for which traditional data processing algorithms and models are no longer suitable. For example, the number of land use classification patches in Yunnan Province investigated in this study is hundreds of thousands at the county level, millions at the city level, and tens of millions at the provincial level. With the development of the social economy and the progress of data acquisition technologies, the number of land use classification patches will continue to increase. Effectively calculating land use change using traditional single-computer calculation models is difficult.

The rise of parallel computing technologies, such as network clustering, grid computing, and distributed processing in recent years has gradually shifted research on high-performance GIS spatial computing from the optimization of algorithms to the parallel transformation and parallel strategy design of GIS spatial computing in a cloud computing environment [6]. Recently, MapReduce and Spark technology have been applied to overlay analysis of massive spatial data, and some results have been achieved. Nevertheless, the massive spatial data is different from the general massive Internet data. The spatial characteristics of spatial data and the complexity of a spatial analysis algorithm determine that simply copying a cloud computing programming paradigm cannot achieve high-performance geographic computing. Therefore, this study chooses the classical Hormann clipping algorithm [7] to analyze and measure the impact of the shape complexity of geographic polygons on parallel overlay analysis, and proposes a Hilbert partition method based on the shape complexity measure to solve the data skew caused by the difference of the shape complexity of polygons. In addition, through the combination of MBR (Minimum Bounding Rectangle) filtering, R-tree spatial index and other optimizations, an efficient parallel overlay analysis algorithm is designed. The experimental analysis shows that the proposed method reduces the number of polygon intersection operations, achieves better load balancing of computing tasks, and greatly improves the parallel efficiency of overlay analysis. When the computational core increases, the algorithm achieves an upward acceleration ratio, and the computational performance presents a nonlinear change.

The rest of this paper is organized as follows. Section 2 reviews the research background and related studies, including those on shape complexity and overlay analysis algorithms. In Section 3, the Hormann algorithm is improved for a parallel polygon clipping process, and the process of a parallel polygon clipping algorithm is optimized according to the shape complexity of polygons. Section 4 describes the experimental process in detail and analyzes the experimental results. Section 5 provides the conclusion drawn from this research, followed by potential future work.

## 2. Relevant Work

This paper discusses the related research work from two aspects: shape complexity and overlay analysis algorithm.

### 2.1. Shape Complexity

Many studies use abstract language to describe the shapes and complex details of geometric objects, such as "the structure of polygons with multiple holes, the number of vertices is very large, and polygons with multiple concaves." To evaluate the computational cost, the complexity and computational efficiency of geometric computing problems should be accurately measured [8]. The concept of complexity is also introduced [9–12]. Many applications related to spatial computing heavily depend on algorithms to solve geometric problems. When dealing with large-scale geographic computing problems, the evaluation of computational cost must consider the quantity of input data, the complexity of graphical objects, and the time complexity of computing models [10]. When the amount of input data and the algorithm are determined, the complexity of different graphical objects frequently leads to considerable differences in the computing efficiency.

Mandelbrot described the complexity of geometric objects from the perspective of a fractal dimension [13]. The most commonly used method is the box-counting technique [14,15]. Brinkhoff

quantitatively reported the complexity of polygons from three aspects, namely, the frequency of local vibration, the amplitude of local vibration, and the deviation from the convex hull, to describe the complexity of a global shape [16]. On the basis of Brinkhoff's research, Bryson proposed a conceptual framework to discuss the query processing-oriented shape complexity measures for spatial objects [17]. Rossignac [8] analyzed shape complexity from the aspects of algebraic, topological, morphological, combinatorial, and expression complexities. Rossignac also reduced the shape complexity by using a triangular boundary representation at different scales [8]. Ying optimized graphic data transmission on the basis of shape complexity [18].

From the above discussion, we know that the complexity of graphics has different meanings and measurement methods in different professional fields, such as design complexity, visual complexity and so on. Therefore, we should consider the shape complexity from the perspective of geographic computing. Shape complexity directly affects the efficiency of spatial analysis and spatial query computation, such as the numbers of vertices and local shapes (such as the concavity) of graphics, considerably influencing the efficiency of spatial geometry calculation. These values are important indicators for evaluating the calculation cost. Fully considering the influence of graphical complexity on specific geographic computations can effectively optimize the computing efficiency of applications.

## 2.2. Overlay Analysis

The study on vector overlay analysis arithmetic originates from the field of computer graphics. For example, two groups of thousands of overlay polygons are often clipped in 2D and 3D graphics rendering. Subsequently, different overlay analysis algorithms have been produced. Among which, the Sutherland–Hodgman [19], Vatti [20], and Greiner–Hormann [7] algorithms are the most representative when dealing with arbitrary polygon clippings. The Sutherland–Hodgman algorithm is unsuitable for complex polygons. The Weiler–Atherton algorithm requires candidate polygons to be arranged clockwise and with no self-intersecting polygons. The Vatti algorithm does not restrict the types of clipping, and thus, self-intersecting and porous polygons can also be processed. The Hormann algorithm clips polygons by judging the entrance and exit of directional lines. This algorithm also addresses point degradation by moving small distances [21]. In addition, the Hormann algorithm can deal with self-intersecting and nonconvex polygons. The Weiler algorithm uses tree data structures, whereas the Vatti and Greiner–Hormann algorithms adopt a bilinear linked list data structure. Therefore, the Vatti and Greiner–Hormann algorithms are better than the Weiler algorithm in terms of complexity and running speed.

Subsequent researchers have implemented many improvements to the aforementioned traditional vector clipping algorithms [22–24], which simplify the calculation of vector polygon clippings. However, these studies are based on the optimization of a serial algorithm. When overlay analysis is applied to the field of geographic computing, it will deal with more complex polygons (such as polygons with holes and islands) and a larger data volume (the number of land use classification patches in a province is tens or even hundreds of millions, and a polygon may have tens of thousands of vertices). A vector clipping algorithm can be applied efficiently to computer graphics but cannot be applied efficiently to geographic computing. Moreover, many traditional geographic element clipping algorithms also exhibit poor suitability and performance degradation. With the development of computer technology and the increase in spatial data volume, traditional vector clipping algorithms frequently encounter efficiency bottlenecks when dealing with large and complex geographic data sets. Therefore, improving the overlay algorithm and using the parallel computing platform for the overlay analysis of massive data is a new research direction.

With the rapid development of MapReduce and Spark cloud computing technologies, the use of large-scale distributed storage and parallel computing technology for massive data processing and analysis has become an effective technical approach [6,25]. Recent studies have applied the MapReduce and Spark technology to the overlay analysis of massive spatial data. Wang [26] used MapReduce to improve the efficiency of overlay analysis by about 10 times by a grid partition and index. Zheng [27]

built a multilevel grid index structure by combining the first-level grid with quartering based on Spark distributed computation platform. Zheng's experiments show that a grid index algorithm achieves good results when polygons are uniformly distributed; otherwise, the efficiency of the algorithm is low. Xiao [28] proves that parallel task partitioning based on polygons' spatial location achieves better load balancing than random task partitioning. In addition, SpatialHadoop [29–32] and GeoSpark [32–35] extend Hadoop and Spark to support massive spatial data computing better. Among them, Spatial Hadoop designed a set of spatial object storage model, which provides HDFS with grid, R-tree, Hilbert curve, Z curve and other indexes. In addition, it also provides a filtering function for filtering data that need not be processed. GeoSpark also adds a set of spatial object models and extends RDD (Resilient Distributed Dataset) to SRDD (Spatial Resilient Distributed Dataset) that supports spatial object storage. GeoSpark also provides filtering functions to filter data that need not be processed. The design ideas of SpatialHadoop and GeoSpark have great reference value for the research of this paper.

In summary, using a Spark cloud computing paradigm to develop high-performance geographic computing is a cheap and high-performance method. It is also one of the research hotspots in the field of high-performance geographic computing. Recent studies have implemented overlay analysis in Spark, which greatly improves the efficiency of overlay analysis. Optimizing strategies for spatial data characteristics, such as reasonable data partitioning and an excellent spatial data index, play an important role in improving the efficiency of parallel computing, and Hilbert partitioning is more suitable for parallel overlay analysis of non-uniform spatial distribution data than grid partitioning. It is noticed that the current parallel overlay analysis is based on the third-party clipping interface and ignores the impact of the shape complexity of geographic polygons on the clipping algorithm, which will cause a serious data skew.

## 3. Methodology

In this section, the core idea of the paper will be introduced. First, an excellent basic overlay analysis algorithm is selected for execution on each computing node. Then, according to the complexity and location of graphics, the polygons are divided reasonably, and an index based on spatial location is established to realize fast data access and load balancing of parallel computing nodes.

### 3.1. Basic Overlay Analysis Algorithm Running on Each Computing Node.

#### 3.1.1. Hormann Algorithm and Improvement of Intersection Degeneration Problem.

The basic overlay analysis algorithm, which is the basic processing program for each parallel computing node, performs overlay analysis on two sets of polygons. In designing an overlay analysis algorithm, we use the Hormann algorithm, which can deal with complex structures (e.g., self-intersection and polymorphism with holes), as reference. However, the point coordinate perturbation method used by the Hormann algorithm is not the best solution for the degradation problem, which brings cumulative errors in the area statistics of massive patches. Therefore, we solve the intersection degeneration by judging the azimuth interval between intersecting correlation lines. We also use the improved algorithm as the basis of parallel overlay analysis.

In order to achieve a simplified expression of the overlay analysis of two groups of polygons, we assume that each group of polygons has only one polygon object, because the overlay analysis of two groups of layers with multiple polygons only increases the number of iterations. The processing steps of the improved Hormann algorithm are as follows:

1. Calculating the intersections of the clipped and target polygons
2. Judging the entry and exit of the intersection point by the vector line segment (judging the entry or exit point of the intersection point) and adding the entry point to the vertex sequence of the clipping result polygon

3. Comparing the azimuth intervals of the degenerated vertices of the intersection points and adding the overlapping vertices of the azimuth intervals to the vertex sequence of the clipping result polygon

4. Forming a new polygon (clipping result) in accordance with the sequence of vertices

As shown in Figure 1a, the clipped polygon $P_1$ and the target polygon $P_2$ intersect. Intersection points $K_1$ and $K_2$ can be obtained through a collinearity equation. By judging the positive and negative values of the product of vector line segments, the intersection points can be judged to enter and exit. As illustrated in the same figure, $\overrightarrow{A_1A_2} \times \overrightarrow{B_1B_2} > 0$, and thus, $K_1$ is the entry point relative to $P_2$. Moreover, $\overrightarrow{A_2A_3} \times \overrightarrow{B_1B_2} < 0$, and thus, $K_2$ is the exit point. The resulting polygon is composed of a sequence of vertices that consist of $K_1$, $A_2$, and $K_2$. As illustrated in Figure 1b–d, the entry and exit points are unsuitable for describing intersection degradation.

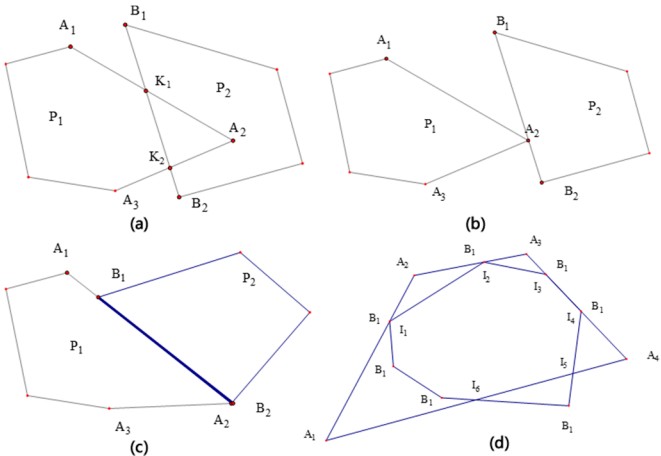

**Figure 1.** Polygon overlay.

Figure 2 shows how to deal with the phenomenon of intersection degeneration. In Figure 2, the dotted arrow $N$ points toward the north, which is the starting point of the azimuth calculation. Therefore, each line segment has its own azimuth. The clipped and target polygons have an intersection point $K$, which is also the location of vertices $A_m$ and $B_m$. The azimuth intervals of the clipped and target polygons at intersection $K$ are $A_C(\alpha_1, \alpha_2)$ and $A_T(\alpha_1, \alpha_2)$. If $A_C(\alpha_1, \alpha_2)$ and $A_T(\alpha_1, \alpha_2)$ have overlapping parts (yellow in the figure), then both polygons overlap near the intersection point, which should be added to the vertex sequence of the resultant polygon.

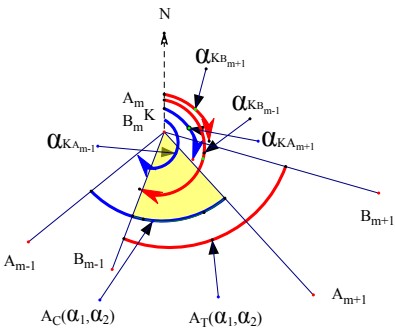

**Figure 2.** Diagram of azimuth interval calculation.

### 3.1.2. Effect of Shape Complexity on Parallel Clipping Efficiency

In parallel clipping computing, each computing node is usually assigned the same number of polygons. Generally speaking, it is difficult to ensure that complex polygons are evenly allocated to

each computing node; usually, one computing node is allocated more complex polygons. Although the total number of polygons allocated by each computing node is the same, this computing node needs a long time to complete the allocated computing task, while other computing nodes will be in a waiting state. Therefore, ignoring the complexity differences of polygons will result in a situation in which each computing node cannot complete the computing task at the same time, thus the efficiency of parallel computing is reduced.

Complexity is an intuitive linguistic concept. Generally speaking, different professional fields pay different attention to shape complexity. In the field of geographic computation, shape complexity is related to specific geographic algorithms. The same shape corresponds to different geographic algorithms and may have different shape complexities.

On the other side, specifically to geographic polygons, different polygons have different morphological characteristics, such as convex, concave, self-intersection and a large number of vertices. To measure complexity, information must be compressed into one or more comparable parameter and expression models. Although the starting point and location are completely different, similar shapes may still appear. Therefore, when discussing the shape complexity of a polygon, we can neglect the spatial position and scale, and focus on the influence of polygon features on geographic computing.

Shape complexity can be defined from the perspective of geographic computing:

**Definition 1.** *Shape complexity is a measure of the computational intensity index of shapes participating in the calculation of geographic algorithms. Shape complexity can be measured by the number of repetitions of basic operations in a geographic algorithm caused by a shape.*

*As for the overlay analysis of polygons, the most basic operation of the Hormann algorithm is to find the intersection point of two sides. Therefore, for the Hormann algorithm, the complexity of a polygon is the number of edges it possesses.*

*Based on these analyses, we know that shape complexity is an absolute value, which is difficult to program. Therefore, it is necessary to get a relative value by normalization to measure the graphics complexity.*

**Definition 2.** *Given a set of polygons $P = \{P_1, P_1, \cdots P_n,\}$, the number of vertices of the polygon is $V_i$, $V_{min}$ is the minimum number of vertices of all polygons, and $V_{max}$ is the maximum number of vertices of all polygons. Then, the complexity $W_i$ of the polygon $P_i$ can be expressed as*

$$W_i = \frac{V_i - V_{min}}{V_{max} - V_{min}} \tag{1}$$

*Since a polygon is usually represented by a sequence of vertices in a polygon storage model. The number of edges of a polygon is the same as that of vertices, so in Definition 2, we use vertices of a polygon instead of edges.*

*Therefore, in parallel overlay analysis, we can take shape complexity as an indicator for data partitioning. The ideal state is that the polygon complexity of each data partition is the same, at which time all computing nodes will complete the computing task at the same time.*

*3.2. Data Balancing and Partitioning Method that Considers Polygon Shape Complexity*

3.2.1. Data Partitioning and Loading Strategy

Data partitioning is the key to accelerating a polygon clipping algorithm based on a high-performance computing platform. A complete piece of data is divided into relatively small, independent multiblock data, which provide a basis for distributed or parallel data operation. Spatial data partitioning differs from general data partitioning. In addition to balancing the amount of data, the spatial location relationship, such as spatial aggregation and proximity of data, should also be considered. Commonly used spatial data partitioning methods are meshing and filling curve

partitioning [36]. Meshing is simple and considers the spatial proximity of data, but it cannot guarantee a balanced amount of data. The Hilbert curve is a classical spatial filling curve with good spatial clustering characteristics and that considers the spatial relationship and data load. Therefore, the data partitioning strategy in this study adopts the Hilbert filling curve algorithm combined with shape complexity to achieve load balancing.

In Figure 3, Hilbert partitioning divides the spatial region into $2N \times 2N$ grids. During the iteration process, $N$ is the order of the Hilbert curve, i.e., the number of iterations. In general, $N$ is determined by the number of spatial objects, and the amount of spatial data requires $n < 2^{2 \times N}$.

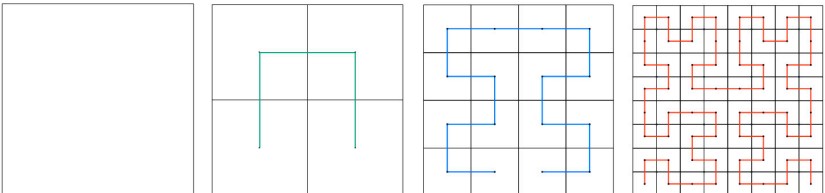

**Figure 3.** Hilbert partitioning and Hilbert curve generation.

Hilbert partitioning consists of the following four steps:

(1) Determine the order of the Hilbert curve, generate the Hilbert grid and the Hilbert curve, number the Hilbert curve sequentially, and obtain the Hilbert grid coding set, GHid $= \{GH_1, GH_2 \cdots GH_n\}$.

(2) Calculate the polygon MBR center point, find its corresponding mesh, and use the Hilbert coding of the mesh as the Hilbert coding of the polygon to obtain the Hilbert coding set of the polygon, PHid $= \{PH_1, PH_2 \cdots PH_n\}$.

(3) In accordance with the number of computing nodes $M$, divide the Hilbert coding set of the polygons into $M$ partitions, and calculate the start–stop coding of the Hilbert coding of polygons in each partition.

(4) Merge the grids of the Hilbert partitions to obtain partition polygons PS $= \{PS_1, PS_2, \cdots, PS_M\}$.

In actual partitioning, the shapes of polygons are different because the polygons are not in an ideal uniform distribution. If only one polygon central point is strictly required for each grid, then Hilbert's order $N$ may be extremely large, the edge length of the grid will be too small, and no polygonal MBR center may exist in many grids. Thus, Hilbert partitioning and the Hilbert curve will consume considerable computing time, and the subsequent overlay calculation will involve many cross-partition problems. Therefore, the existence of multiple polygonal MBR centers in a grid is necessary.

The order $N$ of Hilbert grids is related to the length of the mesh edge. Grid length and order $N$ are also determined. To obtain a reasonable order $N$ of the Hilbert curve, we can calculate the normal distribution of the central point position of a polygon MBR, determine the optimal grid edge length, and eventually achieve balance between the order $N$ of the Hilbert curve and the number of polygon MBR central points in each grid. The key to dividing the PHid of the Hilbert coding set of polygons is to ensure the load balance of each partition. Considering that polygon complexity may vary considerably, we cannot simply divide the Hilbert coding set PHid of polygons equally.

The shape complexity of polygon $P_i$ is defined as $W_i$., $\overline{W}$ as the average complexity of all polygons, the ideal complexity of each partition as $W_{ideal}$, and the actual complexity as $W_{actual}$, then

$$W_{ideal} = \frac{\sum_{i=1}^{n} W_i}{M} \tag{2}$$

if the polygons from j to k are placed into the same partition, then,

$$W_{actual} = \sum_{i=j}^{k} W_i \tag{3}$$

$$|W_{ideal} - W_{actual}| < \overline{W} \tag{4}$$

Generally, the number of polygons in each partition is slightly different after partitioning, but the complexity of polygons in each partition is basically the same. Therefore, this strategy guarantees the load balancing of computing tasks.

### 3.2.2. R-tree Index Construction

R-tree is a widely adopted spatial data index method; it is used in commercial software, such as the Oracle and the SQL Server [37]. To improve the efficiency of spatial data access, an R-tree must be built. In addition, data are segmented in accordance with Hilbert data partitioning points, and the grid area of the Hilbert curve before each partitioning point is defined as a sub-index area. Moreover, the R-tree index for spatial objects is established in the sub-index area. Similarly, the mapping relationship among grid coding, polygon MBR central point coding, and sub-index area coding is established. Furthermore, the corresponding index codes on computing nodes are cached. In this experiment, we directly use the STR-tree (Sort Tile Recursive) class of the JTS (Java Topology Suite, a java software library) library to construct an R-tree index.

### 3.3. Process Design of Distributed Parallel Overlay Analysis

To ensure that the process is suitable for decoupling, we divide the distributed parallel overlay analysis process into six steps based on the characteristics of the algorithm: Data preprocessing, preliminary filtering, Hilbert coding, data partitioning and index building, data filtering, and overlay calculation (Figure 4).

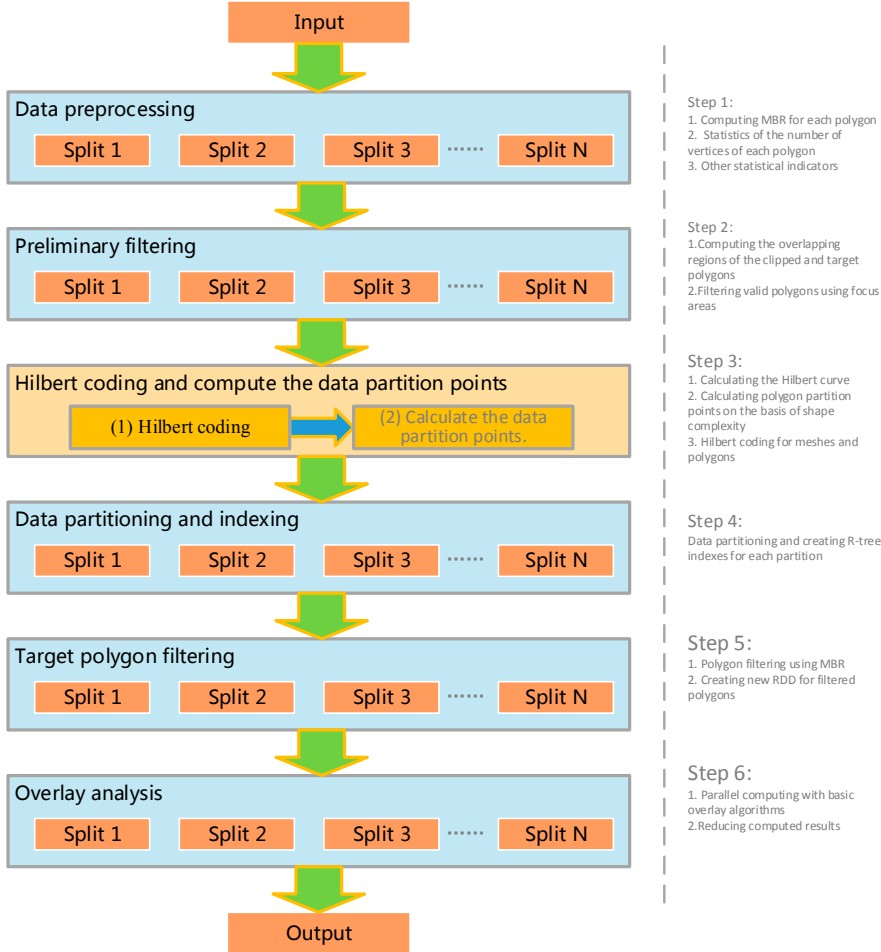

**Figure 4.** Parallel overlay computing flow.

(1)　Data preprocessing

In the whole process, many steps need to traverse all polygons and their vertices. To reduce the number of traversals, we can conduct centralized processing in one traversal, such as calculating the MBR of a polygon, its geometric center point, the area of the MBR, and the shape complexity of the polygon, to prepare data for optimizing the processing flow. In the subsequent calculation, such information can be read directly to avoid repeated calculation. Considering the massive amount of data in preprocessing, the Spark paradigm can be used in parallel data processing. In accordance with the calculation of the number of physical nodes $N$, data are divided by default, and the allocated data are traversed at each computing node.

(2)　Preliminary filtering

It can be determined that only the polygons in the area where the MBRs of two polygonal layers intersect need to be clipped. Therefore, filtering polygons that do not require clipping can reduce the computational cost.

(3)　Hilbert coding and computation of data partition points base on polygon clip complexity

All polygons are divided by Hilbert grids in accordance with the spatial distribution position, and each grid and polygon are Hilbert coded. Then, the data partition points are calculated on the basis of shape complexity. This work is not suitable for decoupling and, therefore, cannot be executed in parallel.

(4)　Data partitioning and indexing

Based on the partitioning points, the region of the Hilbert curve is regarded as a sub-index region. The STR-tree class of the JTS library is used to establish an R-tree index for each partition, and the index file is stored in each computing node.

(5)　Target polygon filtering

In overlay analysis, every point of the clipped and target polygons should be traversed. Even if the two polygons are not covered, all points will be traversed, resulting in some invalid calculations. Filtering out the invalid polygons of target polygons can obviously improve the efficiency. Before overlay calculation, the target polygon without overlay analysis can be effectively eliminated by calculating whether an overlay relationship exists between the MBR of the clipped polygon and the MBR of the target polygon. The calculation method directly compares the maximum and minimum coordinates of the clipped and target polygons without using an overlay algorithm.

(6)　Overlay analysis

All computing nodes use the Hormann algorithm described in Section 3.1 for parallel overlay computation. The results of each calculation node are reduced to obtain the final overlay analysis results.

*3.4. Algorithmic Analysis*

The major processes of the parallel overlay analysis conducted in this study include data preprocessing, preliminary filtering, Hilbert partitioning, R-tree index establishment, polygon MBR filtering, and polygon clipping.

In the data preprocessing, only the layer attribute data and vertex coordinate information of the polygons are included in the original data. Polygon MBR and the number of polygon vertices must be used thrice in the calculation process designed in this research. Therefore, we unified the data preprocessing, establish a new data structure and avoided repetitive calculation. Unified data preprocessing saves about half of the workload compared with separate data preprocessing.

In the Preliminary filtering, the time complexity of the MBR filtering algorithm is O(1), whereas the complexity of the overlay analysis algorithm is O(logN), where $N$ is the number of polygon

vertices. Therefore, computational complexity will be considerably reduced by filtering polygons without overlay analysis through MBR. In addition, the reduced computational complexity depends on the spatial distribution of polygons, which is an uncontrollable factor.

The time complexity of constructing the Hilbert curve is $O(N^2)$, where $N$ is the order of the Hilbert curve. The larger $N$ is, the longer the time that is spent on data partitioning is. However, if $N$ is too small, then multiple polygons will correspond to the same Hilbert coding. If Hilbert partitioning strictly satisfies the condition that each mesh has only one central point of a polygon MBR, then a Hilbert grid supports a maximum of $2^{2 \times N}$ polygons. Moreover, polygons in real data are generally not uniformly distributed, and no polygons exist in many Hilbert grids. Therefore, allowing an appropriate number of repetitive Hilbert-coded values is feasible. In addition, compared with grid partition, Hilbert partition can solve the problem of the uneven location of data perfectly.

R-tree is a typical spatial data index method. The time for data traversal is considerably shortened by establishing an R-tree index. The time complexity of R-tree is $O(logN)$.

Data preprocessing, MBR filtering, R-tree index construction, and other processes are relatively time-consuming. By using multi-node parallel computing in data partitioning, the time consumed can be reduced to $1/N$, where N is the number of parallel processes. In the Spark paradigm, data operations are performed in memory, and I/O operations consume minimal time. Therefore, the proposed overlay algorithm exhibits high efficiency.

## 4. Experimental Study

### 4.1. Experimental Design

To conduct overlay analysis experiments, we used the patches of land use types and the patches with a slope greater than 25 degrees in a county of Yunnan Province in 2018. There are 500,000 patches of land use types and 110,000 slope patches. These data are distributed in the area of 15,000 square kilometers. Based on this data, we constructed different data sets for the experiments. We will use different overlay analysis modes for the execution in the case of different data magnitude data, record the change of execution time, and analyze the characteristics and applicability of different overlay analysis modes.

#### 4.1.1. Computing Equipment

Experiments were carried out using one portable computer and six X86 servers. The equipment configuration information is shown in Table 1.

**Table 1.** Equipment configuration.

| Equipment | Num | Hardware Configuration | Operating System | Software | Remark |
|---|---|---|---|---|---|
| portable computer | 1 | Thinkpad T470p, 8 vcore, 16 G RAM, SSD (Solid State Drive) | Windows 10 | ArcMap 10.4.1 | Single computer experiment for desktop overlay analysis. |
| X86 Server | 6 | DELL R720, 24 core, 64 G RAM, HDD (Hard Disk Drive) | Centos7 | Hadoop 2.7, Spark 2.3.1 | Spark Computing Cluster |

#### 4.1.2. Experimental Data

(1) Clipping layer

The digital elevation model (DEM) data of a 30 m grid in the county were obtained from the Internet, and a slope map was generated by it (Figure 5). The area with a slope greater than 25 degrees was extracted, and 108,025 patches were obtained.

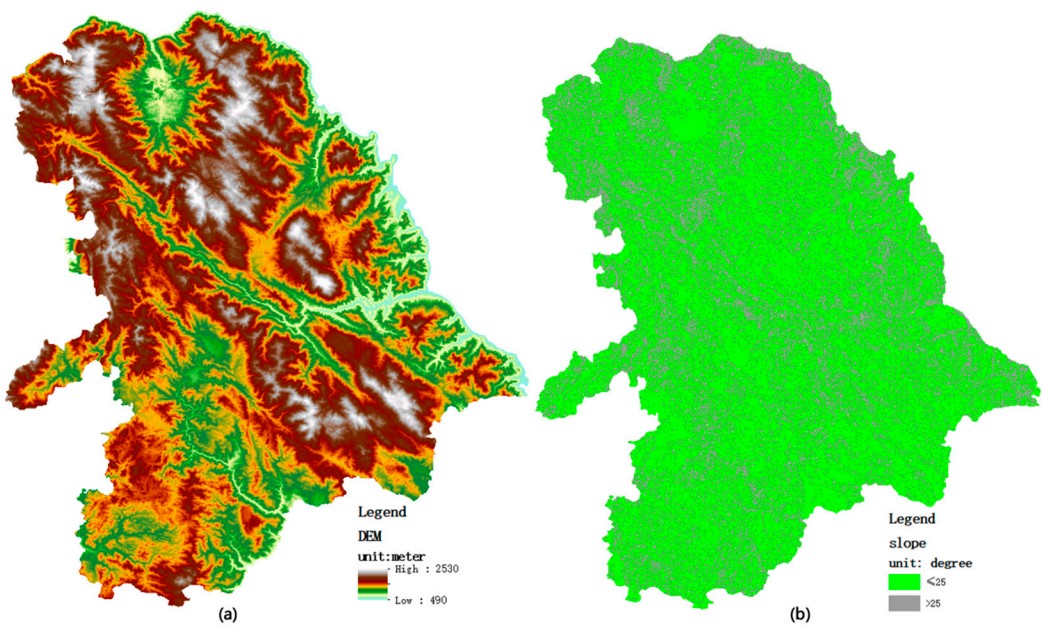

**Figure 5.** Extracting slope as the clipping layer from the digital elevation model (DEM).

(2)  Target layer

A total of 10 groups of experimental data were obtained from 500,000 original land-type patches of the county using sparse and intensive data sets. The number of patches was 50,000, 100,000, 250,000, 500,000, 1 million, 2 million, 4 million, 6 million, 8 million and 10 million, respectively.

Through data checking, 88,000,000 vertices were found in 500,000 original terrain pattern data. Among all the polygons, the simplest polygon has four vertices, whereas the most complex polygon has 99,500 vertices. A total of 890,000 vertices were recorded in 110,000 slope patches. Among all slope patches, the simplest has 8 vertices, whereas the most complex has 5572 vertices. The statistics of the number of polygon vertices in the query and target layers are illustrated in Figures 6 and 7, respectively.

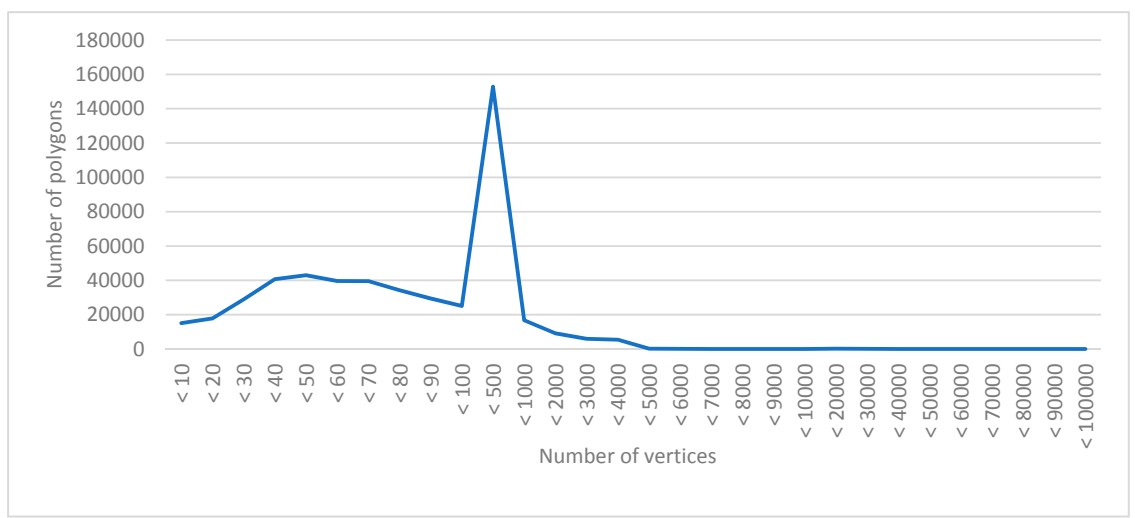

**Figure 6.** Polygons distribution with different number of vertices.

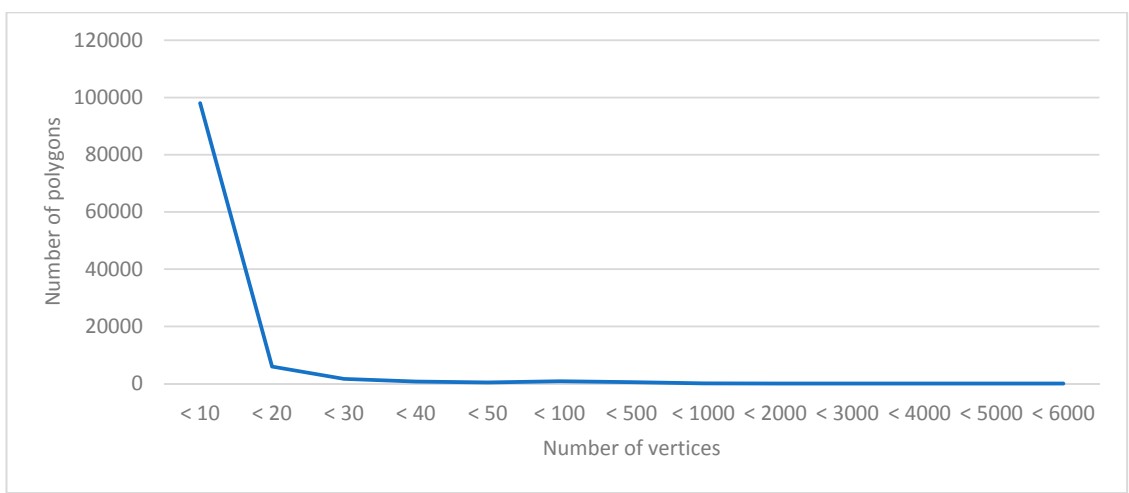

**Figure 7.** Polygons distribution with different number of vertices.

In parallel computing, data are organized into GeoJson format and uploaded to HDFS. HDFS data blocks are three copies, each of which is 64 MB.

### 4.1.3. Experimental Scene

Before describing the experimental scenario design, we first define several different modes for comparison and explain the differences of each mode (Table 2).

**Table 2.** Explanation of the experimental mode.

| Mode Abbreviation | Equipment | Data Storage Mode | Notes |
|---|---|---|---|
| ArcMap | 1 portable computer with ArcMap | Local File System | Use the clip tool of Toolbox to perform overlay analysis on the portable computer |
| Spark_original | Multiple X86 servers with Spark | HDFS | Directly partition the data randomly and do parallel overlay analysis without any improvement. |
| Spark_improved | Multiple X86 servers with Spark | HDFS | Completely implement parallel overlay analysis according to the process of Section 3.3. Hilbert partitioning method considering graph complexity |
| Spark_NoComlexity | Multiple X86 servers with Spark | HDFS | Except that the complexity of polygon graphics is not considered, all of them are the same as the Spark_improved mode. |
| Spark_MBR | Multiple X86 servers with Spark | HDFS | Based on the Spark_original model, MBR filtering is performed first, and then parallel overlay analysis is performed. |
| Spark_MBR_Hilbert | Multiple X86 servers with Spark | HDFS | Based on the Spark_original model, MBR filtering and a Hilbert partitioning operation are added. |
| Spark_MBR_Hilbert_R-tree | Multiple X86 servers with Spark | HDFS | Based on the Spark_original model, MBR filtering, Hilbert partitioning and R-tree index creation operation are added. |

Among them, the ArcMap mode is a typical method used in geographic data processing. The purpose of comparing Spark_original, Spark_improved and Spark_NoComlexity modes is to determine how much the performance has been improved. The purpose of comparing Spark_MBR,

Spark_MBR_Hilbert and Spark_MBR_Hilbert_R-tree modes is to determine how much the three improved methods can improve the efficiency of parallel overlay analysis.

(1) Scene 1: Compare the performance differences of four modes: ArcMap, Spark_original, Spark_improved and Spark_NoComlexity.

Ten groups of polygons with different numbers were used for overlay analysis in four modes. We will record the completion time of the overlay analysis process and draw time-consumption curves. This experimental scenario can answer the following questions:

- How much better will Spark parallel computing improve the performance of overlay analysis compared to desktop software?
- How much better is the performance of the parallel overlay analysis algorithm proposed in this paper compared with the direct use of the spark computing paradigm?
- How much influence does the complexity difference of a geographic polygon have on parallel overlay analysis?

(2) Scene 2: Compare the performance differences of four modes: Spark_original, Spark_MBR, Spark_MBR_Hilbert and Spark_MBR_Hilbert_R-tree.

Ten groups of polygons with different numbers were used for overlay analysis in three modes. We will record the completion time of the overlay analysis process and draw time-consumption curves.

In addition to considering the influence of the shape complexity difference of a geographic polygon, three important improvements are used in our algorithm flow: (1) MBR filtering, (2) Hilbert partitioning, (3) R-tree establishment. This experimental scenario can answer: How much do the above three improvements affect the efficiency of parallel computing?

(3) Scene 3: Cluster acceleration performance testing of the proposed algorithm.

The experimental data are fixed to 10 million geographic polygons. One to six servers are used to perform overlay analysis and record the time-consumption changes of the overlay analysis algorithm in this paper. In this experimental scenario, we can see the acceleration ratio and parallel efficiency of the proposed algorithm in the Spark cluster.

*4.2. Test Process and Results*

4.2.1. Compare the Performance Differences of Four Modes: ArcMap, Spark_original, Spark_NoComlexity and Spark_improved

The parallel computing mode uses six computing nodes to calculate the flow before and after optimization. The experimental data are collected from 50,000, 100,000, 250,000, 500,000, 1 million, 2 million, 4 million, 6 million, 8 million, and 10 million recorded data sets. The time consumption statistics of different computing modes are illustrated in Figure 8.

As shown in Figure 9, blue, red, yellow and grey represent the time consumed by ArcMap, Spark_original, Spark_NoComlexity and Spark_improved modes. With the increase in the data volume, the four time-consumption curves show an upward trend. The changes in these curves answer three questions related to the design of the experimental scenario.

(1) When the number of polygons is less than 10 million, the efficiency of Spark_original mode is even lower than that of ArcMap mode. When the number of polygons is more than 50,000, the time-consumption of the Spark_improved mode is less than that of the ArcMap mode. When the number of polygons exceeds 1 million, ArcMap mode consumes twice as much time as the Spark_improved mode. As the amount of data increases, the time-consumption of the ArcMap mode increases dramatically, and the time-consumption curve of Spark_improved mode is still relatively flat.

(2)   The efficiency of Spark_original mode is lower than that of Spark_improved mode, and the more
polygons there are, the more obvious it is. This shows that the efficiency of overlay analysis using
Spark directly is very low, and the algorithm optimization must be carried out according to the
characteristics of spatial data and geographical calculation.

(3)   By comparing the time-consumption curves, Spark_improved takes almost half as much time as
Spark_NoComlexity, which is better than I thought. I think it may be related to my experimental
data: in Section 4.1.2, I have found that there are many polygons with high shape complexity in
the experimental data. Maybe many big polygons are partitioned into the same computational
partition, which leads to data skew.

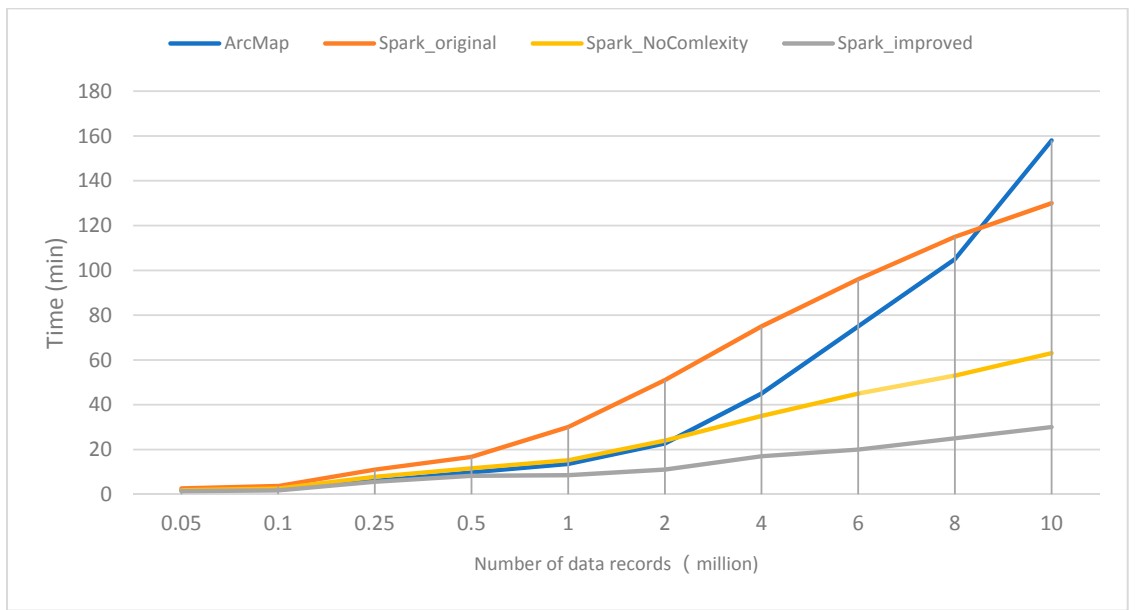

**Figure 8.** Time consumption statistical graphs of different computing modes.

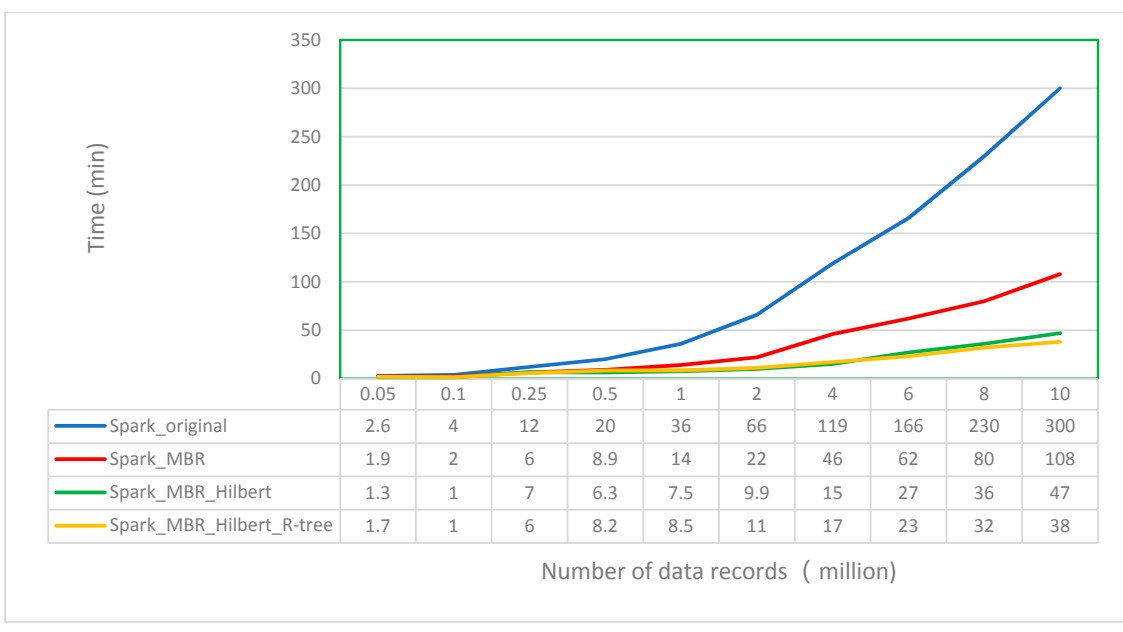

**Figure 9.** Time consumption comparison of different optimization strategies.

4.2.2. Compare the Performance Differences of Four Modes: Spark_original, Spark_MBR, Spark_MBR_Hilbert and Spark_MBR_Hilbert_R-tree

As shown in Figure 9:

(1)  After only adopting the MBR filtering strategy, the efficiency of overlay computation is increased by two to four times. Therefore, this strategy filters a large number of invalid overlay computations. Specific efficiency improvement is related to the size, shape, and spatial distribution of polygons in the target and clipped layers.

(2)  The Hilbert partitioning algorithm based on polygon graphic complexity is used to allocate the data of each computing node. When the amount of data reaches millions, the computing performance can be doubled. As the data amount increases, the computational performance advantage becomes more evident. The experimental data verify that the spatial aggregation characteristics of Hilbert partitioning that considers polygon complexity can considerably improve spatial analysis algorithms.

(3)  Index construction can generally improve the efficiency of data access, but index construction itself can result in a certain amount of computational overhead. After adding the R-tree index strategy based on the first two steps, the overlay calculation time of each order of magnitude increases slightly when the amount of data is less than 5 million. When the amount of data exceeds 5 million, the overlay calculation time decreases compared with the case without the R-tree index. Therefore, the data access time saved after the R-tree index is established offsets the time consumed by the index itself.

4.2.3. Cluster Acceleration Performance Testing of the Proposed Algorithm

The experimental data are unified using 10 million polygons, and then one server is added at a time. As the number of servers increases, the time consumed in parallel computing decreases considerably (Figure 10). However, the trend of running time decreases as the number of nodes increases.

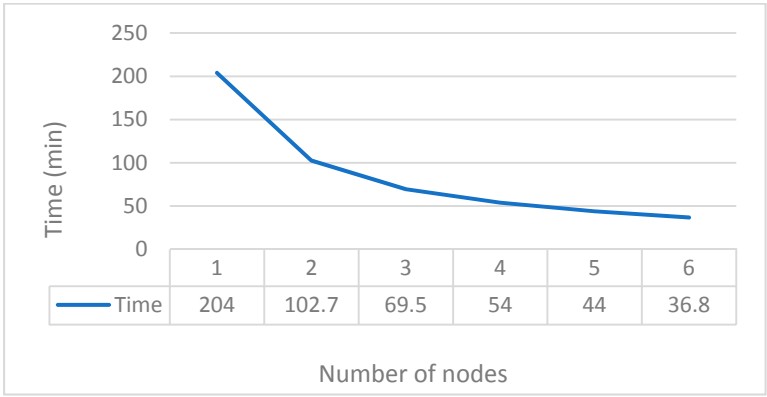

**Figure 10.** Average running time of different numbers of nodes.

As shown in Figure 11, as the number of servers increases, the acceleration ratio decreases slightly but is nearly linear. In addition, Figure 12 illustrates that with the increase in the number of servers, the parallel efficiency gradually decreases, and finally stabilizes to more than 90%. This is a good result if we consider that the increase in the number of servers will increase the system synchronization and network overhead.

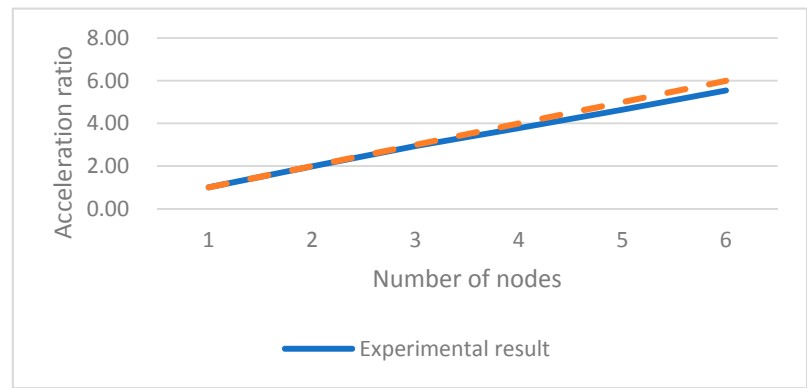

**Figure 11.** Acceleration ratio of different numbers of nodes.

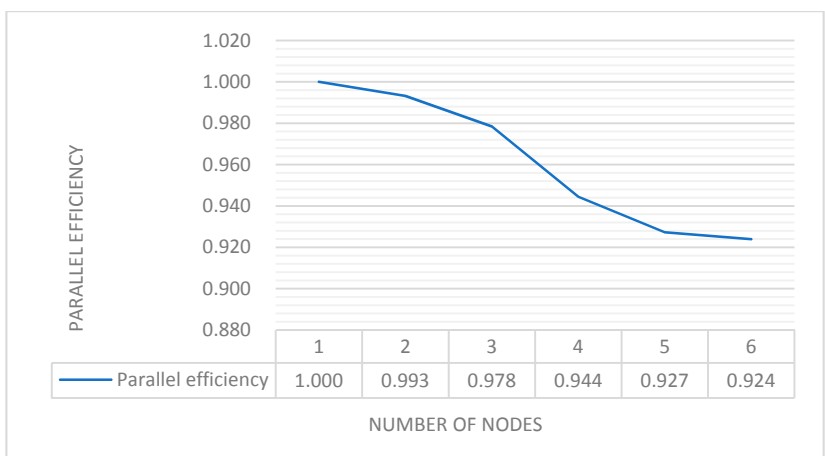

**Figure 12.** Parallel efficiency of different number of nodes.

### 4.3. Analysis of Experimental Results

Figure 9 shows that a single computer with ArcMap Soft achieves high efficiency in the overlay analysis of small data volume by adopting a reasonable algorithm and excellent multithreading processing technology. In addition, SDD also plays an important role. However, the performance of ArcMap sharply declines when the number of data records reaches millions. Spark distributed parallel computing can effectively solve such problems, but the simple transplantation of the overlay analysis algorithm into the Spark framework is not a reasonable solution. The actual geographic data are often unevenly distributed, and the complexity of polygon graphics varies greatly, which will lead to a serious data skew, and which will seriously affect the performance of parallel computing. When the data volume reaches tens of millions, the performance of our algorithm improves by more than 10 times via Hilbert partitioning based on the polygon graphic complexity and the R-tree index. In addition, the performance advantage becomes more evident as the data volume increases.

When the amount of data is constant, the time-consumption of parallel overlay analysis decreases with the increase in the number of servers. However, the decreasing trend of running time declines as the number of nodes increases. Figure 11 shows that the acceleration ratio is nearly linear. Figure 12 illustrates that parallel efficiency is still over 90% and remains stable when the number of servers increases to six, which means that higher computing efficiency can be maintained when the computing cluster expands. Therefore, it is an effective method to add physical nodes in massive data overlay analysis.

In addition, the proposed overlay analysis algorithm also has some problems to be improved, such as: (1) Big polygons will span multiple data partitions, which will lead to repeated participation of the

polygon in overlay analysis on multiple servers. (2) In the current algorithm process, the R-tree index is created temporarily, which leads to the creation of the index repeatedly for each overlay analysis.

## 5. Conclusions

In high-performance parallel overlay analysis, the differences in shape complexity of a polygon can lead to serious data skew. In this paper, we measure the shape complexity of polygons from the perspective of geographic computing and design a high-performance parallel overlay analysis algorithm considering the shape complexity of polygons. The analysis of the algorithm shows that the algorithm reduces invalid overlay calculation by MBR filtering, achieves load balancing by use of a Hilbert partition based on the polygon shape complexity, and improves data access speed using the R-tree index. Experiments show that this is a high-performance method and can maintain high speed-up and parallel efficiency in computing cluster expansion.

In future studies, we will study the impact of the spatial distribution of graphics, spatial data storage and indexing methods on the efficiency of overlay analysis. We will also optimize spatial index storage through distributed memory database technology to further improve the efficiency of parallel overlay analysis.

**Author Contributions:** Kang Zhao proposed the research ideas and technical lines. Baoxuan Jin and Hong Fan provided research guidance. Weiwei Song shared valuable opinions. Sunyu Zhou and Yuanyi Jiang helped complete the programming. Zhao Kang completed the thesis.

**Funding:** This research was funded by Natural Science Foundation of China, grant number 41661086.

**Acknowledgments:** The authors are grateful to Fan Yang and Liying Li for providing their experimental data and to Lifeng Hou for giving valuable suggestions.

**Conflicts of Interest:** The authors declare no conflict of interest.

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
