# Peer review of "High-Performance Overlay Analysis of Massive Geographic Polygons That Considers Shape Complexity in a Cloud Environment"

_ijgi, doi:10.3390/ijgi8070290_

Round 1

Reviewer 1 Report

The paper addresses the problem of overlay analysis, which is a very practical and important problem, in a new setting. The new setting involves two main factors, the availability of large and complex polygons and the advances in distributed computing systems, such as Spark and derivative systems, that are able to provide much better computational power to speed up the overlay queries.

Generally, the paper has two major limitations. First, it is missing very close related work, examples are listed below. It is not clear how the proposed work compares to this existing work. This makes it hard to judge different aspects of the proposed work. Second, the paper presentation needs major improvements to be able to judge the novelty, the technical contributions, and the impact of results. Mainly, the paper should present smoothly the problem definition, the limitations of related work, summary of the proposed contributions in terms of objectives and methods, then details of the methods and results. Major elements of those are missing. For example, the problem definition is not clearly stated. This makes it confusing up to Section 3 to know if the input is two polygons or two sets of polygons, as the later is more practical and expected for distributed environments, while the former is what is discussed in Section 3. Also, the experimental evaluation does not compare with existing techniques.

Examples of missing related work:

* Spark-Based Iterative Spatial Overlay Analysis Method by Z Zheng, C Luo, WU Ye, J Ning in the 2017 International Conference on Electronic Industry and Automation (EIA 2017)

* Interactive and Online Buffer-Overlay Analytics of Large-Scale Spatial Data by M Ma, Y Wu, L Chen, J Li, N Jing in the ISPRS International Journal of Geo-Information

* A vector map overlay algorithm based on distributed queue by Z Xiao, Q Qiu, J Fang, S Cui in the 2017 IEEE International Geoscience and Remote Sensing Symposium (IGARSS)

Other presentations issues:

* Figure 5 is not readable

* Table 2 repeats the same thing again and again

* Figure 4 makes a big disconnection in space

Author Response

Dear Reviewer,

Thank you for your comments. I will think deeply and revise accordingly.Details are described in the attachments. 

Regards,

Kang Zhao

Reviewer 2 Report

Format needs to be modified or consistent in the following:

Line 405, the font size of Figure 10 should be smaller.

Line 481 to 483 should be modified. 

Line 519-521, the format of references should be consistent.

By the way, format in references should be consistent.

Please compare built environment with other tool (such as arcmap of arcgis), such as the speed of ArcMap and proposed distributed parallel processing Spark in overlay analysis.

Author Response

(The authors gave the same response as above.)

Reviewer 3 Report

The article describes an overlay analysis algorithm based on spark paradigm. It is mainly based on the use of a parallel computing  platform  for  massive data processing and analysis. The article is well-organized and sections are well-distributed. However I have some concerns that I think authors must consider:

1) The related works is also well-written and it has a good analysis of the state-of-art in the literature with respect to the three aspects described, graphic complexity, overlay analysis, and parallel computing. However I think that a deep analysis of these areas combined must be added. For example, it is necessary to add a detailed analysis of the  proposals 29-32. In this case, the only sentence about these works is  "Recent  studies  have  applied  the  MapReduce technology  to  the  superposition  analysis  of  massive  polygons [29–32]" without a deep explanation about mainly differences and deficiencies of these works with respect to the one presented here. Even the following sentence affirms that Spark is better than MapReduce without a clear reference in which this fact can be asserted. The reference 33 is only the official Web page of Spark. Where are the advantages of Spark with respect to MapReduce exhibited?

2) I think that for a better understanding of the overlay analysis algorithm proposed is necessary to introduce more explanations in Section 3. This section is composed of subsections 3.1-3.4 (with several more subsections) without an explanation to the connections among them. May be if authors introduce the steps of main parts of the algorithm in a general way (with a figure illustrating it), readers can figure out the idea of the whole process avoiding puzzling the different parts.

3) Similar issues aforementioned can be also applied to Section 4. I think that it is necessary to add basic information in order to understand the way the experiments were performed. Next I enlist some questions a reader commonly look at, in experimental sections (to promote reproducibility): (i) What questions (or hypotheses) are  authors trying to investigate/answer? (ii) How would the goal/s associate with the metrics applied?   (iii) What are the practical implications given the yielded results? (iv) What threats to the validity of the study could be taken into consideration?

For example, some misunderstandings are: (a) are the experiment results of section 4.1.3 (1)  described in subsection 4.2.1? and the experiment results of section 4.1.3 (2) in subsection 4.2.2? (b) which experimental scene is answering the subsection 4.2.3? (c) why are the analysis of the results of each experiment described in section 4.3 for the two last,  and for the first one is on subsection 4.2.1? May be subsection 4.3 must include a more general discussion about the whole experimental scenes in which the reader can see the overall benefits of the algorithm.  

References 1,14,15,24 and 31 must be completed, for example with the year of publication, among other things.

Author Response

(The authors gave the same response as above.)

Round 2

Reviewer 1 Report

According to authors' responses, no revisions actually done. The paper quality is way below the standard.

Author Response

Dear Reviewer,

Generally speaking, I did not fully understand this opinion in the revision of Round 1. After in-depth thinking, I think the main reason is that the structure and ideas of the article are not clear enough, so a lot of revision have been made.Please see the attachment for details.

Regards

Kang Zhao

Reviewer 2 Report

This paper used Spark paradigm to solve the overlay analysis of massive geographic polygons, its calculation efficiency is closely related to factors such as data organization and algorithm design. The authors also designed and implemented a parallel processing algorithm based on Spark paradigm in this paper. It showed that the overlay analysis speed is improved through reasonable data partition, distributed spatial index, minimum boundary rectangular filter and other optimization processes, and the high speed and parallel efficiency are maintained.

This revised paper has added many new updates. The authors should improve the quality of Figure 4 before publication. 

Author Response

Dear Reviewer,

I have made further revisions to the paper, including:

1. I revised Section 2.2 to discuss the improvement of Spatial Hadoop and GeoSpark in large spatial data processing, which refers to six recent references. Although Spatial Hadoop and GeoSpark are not created for overlay analysis, their design ideas have important reference value.

2. I redraw Figure 4 to make the expression of calculation flow more intuitive. At the same time, I revised the text of the process description in order to explain the algorithm more clearly.

3. I deleted Figure 8 which has little value to the paper.

4. Figures 6 and 7 are improved. I have replaced the histogram with the line chart, so that we can more intuitively express the changes in the number of polygons, which is our main concern.

5. I have corrected the grammatical errors. In addition, since I am not a native speaker, I have used a professional English editing service, which helps me correct many grammatical mistakes.

Regards,

Kang Zhao

Reviewer 3 Report

@page { margin: 2cm } p { margin-bottom: 0.25cm; line-height: 120% } a:link { so-language: zxx }

I think that there are still aspects that were not well-addressed, and must be again considered. They are:

1) Respect to the third concern mentioned in the first round review is still incomplete. I suggested authors follow steps defined on some scientific method (based on questions, hypothesis, test, analysis, etc.) [1,2] in order  to promote reproducibility of the study. This point is very important because readers need to know the goals for making the experiments an thus evaluate the quality of them. At the same time, it is important to know if experiments return better results than others works in the literature, such as those described in Related Work section. If this it is not possible, authors must write the reasons.

The same concerns are applied again: (a) are the experiment results of section 4.1.3 (1)  described in subsection 4.2.1? and the experiment results of section 4.1.3 (2) in subsection 4.2.2? (b) which experimental scene is answering the subsection 4.2.3? (c) why are the analysis of the results of each experiment described in section 4.3 for the two last,  and for the first one is on subsection 4.2.1? Section 4 must be  improved in order to show relationships between experimental scene and results.

2) The following sentence, in  the last paragraph of Section 1, must be rewritten: “Section  3: Hormann  algorithm is improved for parallel  polygon clipping process, and the process of parallel polygon clipping algorithm is optimized according to the  polygon complexity.” which is the Hormann  algorithm? It was never aforementioned.   In the previous paragraph authors says that they “improves the traditional overlay algorithm in accordance with the influence of polygon  shape  complexity  on  overlay  analysis”, so Are they modifing  the  Hormann  algorithm?

3) The reference 36 in the related work section (Section 2.3) is still useless.

4) Section 3.3 describes a distributed parallel overlay analysis process into five steps, but Figure 4 has six steps described differently. Step 5 (lines 328-330) is misspelled.

[1]Iris Reinhartz-Berger and Arnon Sturm. 2014. Comprehensibility of UML-based software product line specifications. Empirical Softw. Engg. 19, 3 (June 2014), 678-713. DOI:http://dx.doi.org/10.1007/s10664-012-9234-8

[2]Feldman JA, Sutherland WR. Rejuvenating experimental computer science: a report to the National Science Foundation and others. Communications of the ACM. 1979;22(9):497–502.

Author Response

Dear Reviewer,

Thank you for your valuable comments. I have made corresponding amendments. Please see the attachment for details.

Regards

Kang Zhao

Round 3

Reviewer 1 Report

The paper is way below the quality standard.

Author Response

Dear Reviewer,

I have revised this paper again and used the professional English editing service this time, which helps me correct many grammatical errors. The main modifications include:

1.     I revised Section 2.2 to discuss the improvement of Spatial Hadoop and GeoSpark in large spatial data processing, which refers to six recent references.

2.     I redraw Figure 4 to make the expression of calculation flow more intuitive.

3.     I deleted Figure 8 which has little value to the paper.

4.     Figures 6 and 7 are improved. I have replaced the histogram with the line chart, so that we can more intuitively express the changes in the number of polygons, which is our main concern.

5.     I have corrected the grammatical errors. At the same time, Since I am not a native speaker, I have used a professional English editing service, which helps me correct many grammatical mistakes.

Regards,

Kang Zhao

Reviewer 3 Report

I think that now authors have addressed all my concerns adequately. My only that concern is about figure 4; i think that it must be redesigned in a different way in order to represent the same as it is written in the text. Also it is blurred and a little confused.

Author Response

Dear Reviewer,

I have revised the paper carefully again. The main improvements include:

1.     I redraw Figure 4 to make the expression of calculation flow more intuitive.

2.     I deleted Figure 8 on your recommendation.

3.     Figures 6 and 7 are improved. I have replaced the histogram with the line chart, so that we can more intuitively express the changes in the number of polygons, which is our main concern.

4.     I revised Section 2.2 to discuss the improvement of Spatial Hadoop and GeoSpark in large spatial data processing, which refers to six recent references.

5.     I have corrected the grammatical errors you listed. At the same time, Since I am not a native speaker, I have used a professional English editing service, which helps me correct many grammatical mistakes.

Regards,

Kang Zhao
